# Effect of Physical Activity on Static and Dynamic Postural Balance in Women Treated for Breast Cancer: A Systematic Review

**DOI:** 10.3390/ijerph20043722

**Published:** 2023-02-20

**Authors:** Aleksandra Bula, Karolina Tatar, Regina Wysocka, Kasper Chyrek, Laura Piejko, Agnieszka Nawrat-Szołtysik, Anna Polak

**Affiliations:** 1Institute of Sport Sciences, Department of Physical Therapy, The Academy of Physical Education, 40-065 Katowice, Poland; 2Institute of Physiotherapy and Health Sciences, Department of Physical Therapy, The Academy of Physical Education, 40-065 Katowice, Poland; 3Student Scientific Association, The Academy of Physical Education, 40-065 Katowice, Poland; 4Tommed Medical and Rehabilitation Center, 40-662 Katowice, Poland; 5Doctoral School, Academy of Physical Education, 40-065 Katowice, Poland; 6Clinical Department of Physiotherapy in Mental Diseases of the Academy of Physical Education, Psychiatric Hospital, 40-200 Rybnik, Poland

**Keywords:** breast cancer, physical activity, physical exercise, postural balance, stability training

## Abstract

Background: Therapies against breast cancer (BC) frequently involve complications that impair patients’ daily function and quality of life, the most common of which are motor coordination and balance disorders, increasing the risk of falls and injuries. In such cases, physical activity is recommended. Designed following the PRISMA guidelines, this study presents a systematic review of randomised and pilot clinical trials investigating the effect of physical exercises on postural balance in women treated for BC. Methods: Scientific databases (PubMed, EBSCO) and the online resources of grey publications were searched for trial reports published between January 2002 and February 2022. The inclusion criteria necessitated full-text, English-language reports from randomised clinical trials (RCTs) or pilot clinical trials (pilot CTs), whose authors used physical exercises to treat women with BC and the experimental and control groups consisted of at least 10 women. The methodological quality of the RCTs and pilot CTs were measured using the Physiotherapy Evidence Database (PEDro) scale and the Methodological Index for Non-Randomized Studies (MINORS), respectively. Data were extracted on the effect of exercise on the women’s static and dynamic balance. Results: Seven reports, five RCTs and two pilot CTs involving a total of 575 women (aged 18–83 years) were included in the systematic review. Their training protocols utilised a variety of aerobic, strength, endurance, sensorimotor, Pilates exercises, and fitness exercises with elements of soccer. The experimental groups usually worked out in fitness or rehabilitation centres under the supervision of physiotherapists or trainers. Training sessions of 30–150 min were held 2 or 3 times a week for 1.5–24 months. Most trials reported that static and dynamic balance in the experimental groups improved significantly more compared with the control groups. Conclusions: Physical exercises are able to improve static and dynamic postural balance in women treated for BC. However, as all evidence in support of this conclusion comes from only two pilot CT and five RCTs whose methodologies varied widely, more high quality research is needed to validate their findings and determine which exercise protocols are the most effective in improving postural control in women with BC.

## 1. Introduction

With an incidence rate exceeding 29%, breast cancer (BC) is amongst the most common cancers affecting women in industrialised countries [1]. The epidemiological data show that 226,870 women in the United States of America (USA) and more than 70,000 in Germany had a positive diagnosis for BC in 2012 and 2013–2014, respectively [1,2]. In the USA, BC accounted for 81% of malignant tumours diagnosed in women aged 50 years and older from 2009 through to 2013, of whom only 11% survived [3]. New and enhanced therapies against BC and the rising awareness of the importance of cancer screening are resulting in more and more women with BC being treated early enough to have a good chance of recovery.

Breast cancer stage is assessed based on tumour size, location, and the degree of involvement of the lymph nodes. Starting with the least severe, BC can be classified as stage I, IIa, IIb, IIIa, IIIb, IIIc, and IV [4]. Depending on the BC stage, local therapies (including surgery and radiotherapy) and/or systemic therapies (hormonal therapy, chemotherapy, and immunotherapy) are prescribed [5].

Breast cancer therapies frequently lead to complications that hinder the recovery of patients and affect their quality of life. One serious complication is chemotherapy-induced peripheral neuropathy (CIPN), manifested through limb pain, muscular feebleness, chronic fatigue, and motor and balance disorders [6]. Breast cancer surgery may result in tissue necrosis and scarring at the operated site reducing spine, chest, shoulder girdle, and upper limb mobility, weakening the torso and upper limb muscles, and disturbing biomechanical and postural control [7]. Radiotherapy and hormonal therapy can precipitate menopause and cause hormonal imbalance contributing to bone mass reduction [8,9]. Immunotherapy is indicated to be associated with the inflammatory diseases of internal organs, chronic fatigue, and sometimes with adverse changes in the central nervous system [10]. Complications accompanying BC therapy, mainly impaired neuromuscular function, reduced muscle strength and exercise tolerance, adversely affect mobility and postural control and increase female patients’ risk of falls and injuries [11,12,13].

Many of these complications can be attenuated by appropriately designed physical exercise programmes. Research has shown that as well as alleviating CIPN symptoms and improving the quality of life of female BC survivors [14,15,16,17,18], physical activity can also increase their muscle strength and endurance and improve motor coordination and postural control [7,18,19,20,21].

### Aim of Study

This systematic review of clinical trials (CTs) was undertaken in order to gather evidence-based knowledge about the benefits of physical exercise for static and dynamic balance in women treated for BC.

## 2. Methods

### 2.1. Protocol and Ethics Proclamations

This systematic review was designed and conducted following the Preferred Reporting Items for Systematic Reviews and Meta-Analysis (PRISMA) guidelines [22] (PROSPERO Register code: CRD42023390014). The clinical trials included in the review were conducted as per the Declaration of Helsinki and were approved by relevant ethics committees. All patients signed informed consent forms.

### 2.2. Eligibility Criteria

Trials were included in the review if: (1) they were randomized clinical trials (RCTs) with female BC survivors and had a full-text report in the English language, (2) the authors formed at least one experimental group (EG) consisting of at least 10 women who regularly performed physical exercises (at least some of which was supervised) and compared its results with a control group (CG) who exercised unsupervised or did not exercise beyond their usual daily living activities, (3) the authors assessed the effect of physical exercises on static and or dynamic body balance, (4) the authors formed several EGs performing different physical exercises, and (5) they were pilot CTs assessing the effect of physical exercise on postural control in female BC survivors, consisting of at least 10 women who regularly performed physical exercises.

### 2.3. Search Strategy and Selection Process

Between March 2022 and June 2022, two of the review authors (AB, KT) independently searched electronic scientific databases (PubMed, EBSCO) using keywords such as “breast cancer” AND “physical activity” OR “physical exercise” OR “exercise training” AND “balance” OR “postural balance” OR “fall risk”. The specific search strategies for the different databases are shown in Table 1. The same authors (AB and KT) examined the reference lists of the identified records to find articles published outside of the PubMed and EBSCO database (in the so-called grey zone) that met the review inclusion criteria. In cases when online access to the publications was not possible, two of the review authors (AB and KT) reached out to their authors to obtain the full versions.

The titles, abstracts, and full texts of the selected articles were then independently examined by four authors (AB, KT, RW, and KC), who selected RCTs and CTs according to the predefined inclusion criteria. Any doubts as to the eligibility of particular articles were settled by the fifth author (AP).

### 2.4. Data Collection Process

The selected trials’ purpose, design, patient inclusion and exclusion criteria, research methods, findings, and conclusions were examined by three authors (AB, KT, and AP). The findings of this systematic review were read and approved by all authors.

### 2.5. Data Items

The review concentrated on changes in static and dynamic body balance in female BC patients who regularly performed physical exercises in EGs and those in CGs who did not exercise or exercised unsupervised.

Physical exercise programmes performed by the EGs were thoroughly analysed, as well as the methods used by the trials’ authors to assess participants’ body balance and the basic characteristics of EGs and CGs, such as size, patients’ age, BC stage, the percentages of women who had undergone conservative surgery and mastectomy, and the percentages of women who had received chemotherapy, radiotherapy, immunotherapy, or hormone therapy after surgery.

### 2.6. Methodological Quality Assessment of Trials

The methodological quality of the RCTs and pilot CTs was rated independently by 2 authors (AB, LP) by means of the 10-item Physiotherapy Evidence Database (PEDro) scale [23,24] and the 16-item Methodological Index for Non-Randomized Studies (MINORS) [25]. Discrepant scores were resolved by a third author (AN-S).

### 2.7. Synthesis Methods

Three of the review authors (AB, KT, and AP) synthesised the data reported by particular trials according to their purpose, design, patient inclusion and exclusion criteria, research methods, findings, limitations, and conclusions. The findings of this systematic review were read and approved by all authors.

The trial reports were summarised to highlight the intervention methods, measurement methods, results, limitations, and conclusions. The data used to evaluate the methodological aspects of the trials, groups’ characteristics, physical training methods, methods used to assess intervention impacts, and the trials’ results are synthetically presented in tables.

## 3. Results

### 3.1. Study Selection and Study Characteristics

The searching of the scientific databases with a combination of keywords produced 296 articles. Of those, three were rejected as duplicates and twenty-one were left out because their full texts were not published in the English language. The analysis of the remaining 272 articles showed that 211 of them were neither RCTs nor pilot CTs, so these, too, were removed from the set. Thus, 61 articles remained, of which 54 were disqualified because they were trial protocols (*n* = 3), concerned trials with groups containing other subjects in addition to women with BC (*n* = 8), had EGs that did not exercise regularly or exercised unsupervised (*n* = 26), or did not directly assess the effect of physical exercises on static and dynamic body balance (*n* = 17).

As a result, seven reports were included in the review: five RCTs [18,26,27,28,29] and two pilot CTs [30,31]. The process of searching and selecting trials is outlined in Figure 1.

### 3.2. RCT Quality Assessment

The methodological quality of the five RCTs was assessed on the PEDro scale using the following score ranges: low quality: 1–4 pts (two trials [18,28]); medium quality: 5–7 pts (three trials [26,27,29]), and high quality: 8–9 pts (zero trials) (Table 2).

The methodological quality of the pilot CTs [30,31] was assessed with the MINORS using the following score ranges: low quality, 1–6 pts; medium quality, 7–11 pts; and high quality, 12–16 pts. Both pilot CTs were a found to be low quality studies (6 pts) (Table 3).

RCTs—randomized clinical trials; Criteria: (1) subjects were randomly allocated to groups; (2) allocation was concealed; (3) the groups were similar at baseline regarding the most important prognostic indicators; (4) there was blinding of all subjects; (5) there was blinding of all therapists who administered the therapy; (6) there was blinding of all assessors who measured at least one key outcome; (7) measures of at least one key outcome were obtained from more than 85% of the subjects initially allocated to groups; (8) all subjects for whom outcome measures were available received the treatment or control condition as allocated or, where this was not the case, data for at least one key outcome were analysed by “intention to treat”; (9) the results of between-group statistical comparisons were reported for at least one key outcome; and (10) the study provided both point measures and measures of variability for at least one key outcome.

### 3.3. Group Characteristics

The authors of the five RCTs [18,26,27,28,29] and two pilot CTs [30,31] recruited 575 women aged 18 to 83 years. The RCTs had 467 participants [18,26,27,28,29], of whom 243 were allocated to EGs and performed various physical exercises, and 224 were controls who exercised unsupervised [18,27] or whose only exercise was normal daily activities [26,28,29]. The stage of patients’ BC (0-III) was only provided in three of the RCTs [26,27,28]. All RCT participants had undergone breast-conserving surgery or mastectomy, but the percentages of women who received either treatment were only specified in two trials [28,29]. In all trials [18,26,27,28,29], from 60.4 to 100% of the participants had also received chemotherapy post-surgery. The authors of four trials [26,27,28,29] included information that 45.3–82.8% of the trial participants had been given radiotherapy after surgery, and one RCT [18] did not make any reference to radiotherapy as an additional treatment. Only two RCTs [27,28] indicated the percentages of women who had undergone immunotherapy before enrolment (15.1% and 30.9%, respectively). The percentage of women who received hormone therapy before the study was determined in three RCTs [27,28,29] and was, respectively, 41.5%, 79.4%, and 39.5%. The characteristics of RCT participants are summarised in Table 4.

The pilot CTs [30,31], neither of which had a CG, recruited 108 women. The authors of one of them [30] enrolled women with Stage I–III BC but did not state whether and how many of them had been treated with conservative surgery or mastectomy. They indicated, however, that 67.9% of the participants had undergone both chemotherapy and radiotherapy, and 32% had received chemotherapy or radiotherapy. Whether any of them had also received immunotherapy and hormone therapy was not stated. In the other pilot CT [31], all women had undergone surgery for BC, but neither BC stages nor the type of surgery (breast-conserving or mastectomy) was provided. The authors of the trial [31] indicated, however, that 85%, 75%, and 39.5% of the women, respectively, received chemotherapy, hormone therapy, and radiotherapy post-surgery. The characteristics of the pilot CT participants are provided in Table 5.

### 3.4. Measures

Participants’ dynamic balance was evaluated in all RCTs [18,26,27,28,29] and pilot CTs [30,31] and static balance in both pilot CTs [30,31] and three RCTs [18,27,28]. The authors of the trials adopted different measurement methods.

All trials [18,26,27,28,29,30,31] used functional tests to make dynamic balance assessments (the timed backward tandem walk test; the short physical performance battery; the mini balance evaluation systems test; the functional reach test; the timed get up and go test, and the tandem walk test).

There was only one RCT [18] where static balance assessments were made using a posturometry; the other two RCTs [27,28] and both pilot CTs [30,31] used functional tests, namely the one-leg stance test [27,30,31] and the flamingo balance test [28]. In one trial [18], participants’ static and dynamic balance was additionally evaluated by means of the Fullerton advanced balance scale.

The methods of postural balance assessment used in the RCTs and pilot CTs are described in Table 6.

Additionally, the authors of the trials assessed the muscle strength of the participants’ lower limbs [26,27,28,31], lower body [18,30], upper limbs [18,26,27], and chest [27,31] and evaluated them for the level of experienced fatigue [18,27], fall incidence [26], exercise tolerance [31], physical function [27], the level and types of physical activity [27,30], the degree of disability [27], quality of life [19], severity of neuropathic symptoms [28], bone density [28], fat percentage and lean mass [28], bone turnover markers [28], body posture [29], upper limb flexibility [31], gait speed [30], and endurance of lower limb extensors and flexors [30].

### 3.5. Methods of Physical Activity

The authors of the trials differed in the choice of physical exercise methods. Two RCTs [26,27] used different protocols of strength exercise; regarding the other three RCTs [18,28,29], one opted for endurance and sensorimotor exercises [18], one for fitness exercises with elements of soccer [28], and one for Pilates exercises [29].

In three RCTs [18,28,29], all exercise sessions were conducted in therapeutic centres and were supervised. Regarding the remaining two RCTs [26,27], some exercise sessions were supervised and others were performed by patients at home. Twiss et al. [26] divided their training programme between home-based exercises that the participants performed for the first 8 months of the intervention and training sessions delivered at fitness clubs for the remaining 16 months. In the trial by Winters-Stone et al. [27], the participants exercised twice weekly at a therapeutic centre and once weekly at home.

The authors of pilot CTs [30,31] chose multimodal training with aerobic, resistance, balance, and flexibility exercises that the participants performed at a therapeutic centre [30] or aerobic, resistance, and balance exercises that were supervised for the first 1.5 months and performed by patients alone at home for the next 1.5 months [31].

In all RCTs, interventions ranged in duration from 4 months to 24 months (specifically, 4 [29], 4.5 [18], 12 [27,28], and 24 [26] months), and the participants exercised two or three times weekly for 30–60 min [18,26,27,28,29]. In the two pilot CTs, participants exercised once to twice weekly for 90—150 min for 3 months [30,31].

### 3.6. Methods and Results of RCTs

The methods and the results regarding changes in body balance after physical exercise in the RCTs are presented in Table 7.

Twiss et al. [26] (2009; a PEDro score of 5) conducted their RCT with 223 post-menopausal BC survivors (at a mean age of 58.69 ± 7.5 years and with a mean time since menopause of 7.52 ± 7.5 years), who had completed BC treatment (surgery, radiation therapy, or chemotherapy) at least 6 months before the trial (5.95 ± 6.1 years on average). The study participants were randomised into an EG (*n* = 113) or a CG (*n* = 110). In addition to vitamin D, calcium, and risedronate sodium that both groups were supplemented with, the EG additionally performed physical exercises strengthening the hip girdle, back, and lower and upper limb muscles under the supervision of a coach. The control group only continued their everyday activities. In the first eight months, they performed recommended exercises twice weekly for 30–45 min; in the following 16 months, they worked out twice weekly at fitness clubs. The home workout involved exercises with loads below 20 pounds (1 pound = 0.453 kg); the loads applied during the fitness club sessions were not stated [26]. Adherence to exercises was measured using self-report of number of prescribed sessions attended and participants’ reports of falls. Mean adherence over 24 months was 69.4%. Mean 24-month adherence to exercises was 69.4%. A total of 50 of 110 women attended exercise sessions more than 80% of the time. Mean percentage adherence for 8 months of home-based exercises was 79.7% and for 16 months of fitness centre exercises, it was 60.6%. The primary outcome of the study was 24-month changes in participants’ functional dynamic balance and gait efficiency (the timed backward tandem walk test). The 24-month gains in balance control (*p* = 0.010) proved to be statistically significantly greater in the EG than in the CG (Table 7). A limitation of the trial reported by the authors was both groups taking a daily dose of 500–600 IU of vitamin D. They indicated, however, that the effect of vitamin D on the groups’ results was unlikely to be significant, if it occurred at all, referring to the estimation by Heaney [32] that post-menopausal women should take 1000–2000 IU of vitamin D daily. They also stated that participants’ reporting by phone on their compliance with the regime of home exercises for the first 8 months of the intervention did not compromise the authors’ full control over how participants performed them [26].

The second RCT was conducted by Winters-Stone et al. [27] (2012; a PEDro score of 7), who also recruited post-menopausal BC survivors who had completed chemotherapy and radiotherapy ≥ 1 year before. A group of 106 women was randomised into an EG (*n* = 52; a mean age of 62.3 ± 6.7 years) or a CG (*n* = 54; 62.2 ± 6.7 years). The CG only performed low-intensity stretching exercises, whereas the EG carried out resistance exercises with loads between 60 and 80% of the one-repetition maximum (1-RM). Both groups exercised three times weekly for 60 min for 12 months. A training session consisted of 1–3 sets of resistance exercises selected to strengthen participants’ lower limbs, abdominal, chest, hip girdle, and back muscles, which were performed in a series of 8–12 repetitions with 1–2 min rest between the sets. Winters-Stone et al. [27] progressively increased training loads over the intervention period. Their appropriateness was established by measuring the study participants’ 1-RM once monthly in the first 3 months and then every 2 months until the end of the intervention. For women who could achieve more than 12 repetitions, loads were increased to reduce the number of repetitions to 8. Both groups were assessed at baseline and after 6 and 12 months for functional static balance (the one-leg stance test), and functional dynamic balance and gait efficiency (the short physical performance battery). The authors did not find an improvement in functional static body balance nor in functional dynamic balance and gait efficiency in the EG to be statistically significantly greater than in the CG (*p* > 0.05) (Table 7). A limitation of the study was small group sizes because of insufficient availability of potential participants. Some BC survivors, particularly older ones, could not participate in the trial due to poor mental health and problems with performing even daily activities that they experienced even one year after cancer therapy. Additionally, the authors of the trial did not assess the long-term results of the intervention.

Vollmers et al. [18] (2018; a PEDro score of 3) conducted their study with 36 women undergoing chemotherapy for BC and diagnosed with CIPN. They were randomly assigned to a CG (*n* = 19; a mean age of 52.39 ± 10.14 years) and an EG (*n* = 17; a mean age of 48.56 ± 11.94). The CG was only informed about the importance of staying physically active and was recommended types of physical activity, whereas the EG attended two weekly sessions of physical exercise consisting of a warm-up and exercises strengthening participants’ upper and lower limb muscles (six exercises, each with twenty repetitions), and sensorimotor exercises improving their static and dynamic balance. The intervention lasted 18 weeks, with its first 12 weeks overlapping with chemotherapy. Exercise intensity was set to be high but not exhaustive (13–15 on the 20-point Borg scale). The participants were assessed for functional static and dynamic balance (the Fullerton advanced balance scale) and for static balance with use of posturometry. The post-intervention measurements showed statistically significantly greater improvement in functional static and dynamic balance in the EG compared with the CG (*p* = 0.004) (Table 7). Statistically significantly smaller sway area in monopedal stance (data in cm^2^; assessed with use of posturometry) for both the left leg (*p* = 0.003) and the right leg (*p* = 0.01) in the EG participants obtained after 12 weeks (at the end of chemotherapy) suggested a statistically significantly greater improvement in their static balance compared with the CG (Table 7). The main limitation of the trial reported by its authors was that its long-term results were not assessed.

Uth et al. [28] (2021; a PEDro score of 4) recruited women who had completed chemotherapy and radiotherapy for early stage BC (*n* = 68) at least 5 years before and randomised them into a CG (mean age 50 ± 9.3 years; *n* = 46), participants in which were asked to continue their daily physical activities, and an EG (mean age 47.4 ± 9.4 years; *n* = 22), participants in which were subjected to fitness training with elements of soccer. Training sessions lasting between 45 and 60 min were delivered twice weekly for 12 months and consisted of a dynamic warm-up, passing a ball in pairs, and a game of soccer played on a small pitch. The participants were assessed for functional static balance (the flamingo balance test) at three time points, i.e., before the intervention and after 6 and 12 months of exercising. After 6 months of the intervention, the values of the investigated variables were not statistically significantly different between the EG and the CG, but the post-intervention measurements showed that functional static balance improved significantly more in the EG (*p* = 0.021) (Table 7). Regarding the trial’s limitations, its authors pointed to participants being on average younger than most women treated for BC, the lack of information about their menstrual status (oestrogen concentrations are associated with bone density), and the asymmetry of information regarding fall-related injuries in both groups (the EG women reported these throughout the intervention period, whereas those in the CG were asked about them only after the trial ended). Uth et al. [28] did not verify the long-term results of their intervention.

De Bem Fretta et al. [29] (2021; a PEDro score of 5) recruited 34 female BC survivors after hormonal therapy for their RCT and randomly assigned them to an EG (a mean age of 53.33 ± 8.58 years; *n* = 18) or a CG (a mean age of 57.5 ± 13.02 years; *n* = 16). Both groups were instructed on avoiding lymphedema and asked to continue their normal daily living activities. The intervention in the EG consisted of a programme of Pilates exercises that the participants performed under the supervision of a physiotherapist three times weekly for 60 min for 16 weeks. An exercise session consisted of a 10 min warm-up, a forty-minute bout of Pilates exercises, and a 10 min bout of relaxation exercises concluding a session. Functional dynamic balance (the mini balance evaluation systems test) was assessed two weeks before the RCT and after it ended. Post-intervention measurements showed a statistically significantly greater improvement in functional dynamic balance in the EG compared with the CG (*p* = 0.034) (Table 7). The limitations of the trial, as reported by its authors, included small group sizes and the EG completing only 68% of the training programme (due to missed sessions or some exercises being performed carelessly or not at all).

### 3.7. Results of Pilot CTs

The methods and the results regarding changes in body balance after physical exercise in the pilot CTs are presented in Table 8.

Foley et al. [30] (2016; a MINORS score of 6) recruited 52 women (a mean age of 59.7 ± 10.4 years), who had received BC treatment (surgery, chemotherapy, and/or radiotherapy) 4.96 ± 6.3 years before. All women carried out a multimodal training programme (encompassing aerobic, resistance, balance, and flexibility exercises) supervised by physiotherapists, which was delivered at a therapeutic centre. Ninety-minute exercise sessions were held twice weekly for 12 weeks. During the first 2 weeks, the participants performed aerobic exercises such as a 10–20 min treadmill walk with an intensity of 70–85% of the maximum heart rate. Its duration was progressively extended over the next 10 weeks to 30 min in week 12. Resistance training consisted of 1–2 sets of dynamic (concentric and eccentric) exercises with 8 to 12 repetitions performed at 60–70% 1-RM resistance. Exercise intensity was increased by 5–10% as the women became capable of performing 12 repetitions. Dynamic and static balance and flexibility training comprised exercises on a ball, ball tossing, reaching for objects, bends, dance exercises, stretching yoga poses, and deep diaphragmatic breathing exercises. Participants’ functional static balance (the one-leg stance test), functional reach and dynamic balance (the functional reach test), and functional gait efficiency and dynamic balance (the timed get up and go test) were assessed pre- and post-intervention. After the therapy, the functional static and dynamic body balance improved statistically significantly in all tests (*p* < 0.001) (Table 8). The main limitations of the trial indicated by the authors were the absence of a control group and the non-assessment of the intervention’s long-term results.

The pilot CT conducted by Lee et al. [31] (2016; a MINORS score of 6) involved 56 female BC survivors at a mean age of 53.8 ± 9.6 years who had completed treatment at least one month before the trial. The participants attended six supervised sessions delivered at a Breast Health Centre, which contained a theoretical (educational) component and a practical component. The practical component was aimed to familiarise the participants with types of physical exercises and instruct them how to perform them at home. The home-based exercise programme consisted of (1) aerobic exercise (150 min of a moderate-intensity walk followed by gentle stretching of the lower limbs) performed once weekly; (2) resistance exercises (1–3 sets of progressive exercises with weights or Theraband^TM^, each repeated 8–12 times) performed two or three times weekly to strengthen the trunk and upper and lower limb muscles, and these sessions concluded with two sets of upper and lower limb stretching exercises; and (3) balance exercises (one-leg standing for up to 1 min for each leg, two repetitions for one side), performed twice or three times weekly. Participants exercised once weekly for 150 min for 6 consecutive weeks. Participants’ functional static and dynamic balance (with The The One-Leg Stance Test and The Tandem Walk Test, respectively) were assessed 6 weeks before the trial and pre- and post-intervention. At post-intervention (week 6), statistically significant improvements (*p* < 0.05) were recorded in participants’ static and dynamic balance (Table 8). The main limitations of the trials included the absence of a control group, the relatively short duration of the intervention (6 weeks only), and the non-assessment of its long-term results.

## 4. Discussion

Our systematic review only found five RCTs and two pilot CTs, whose authors evaluated the effect of physical exercise on static and dynamic body balance in women treated for BC. The methodological quality of the five RCTs measured with the 10-point PEDro scale ranged between 3 and 7 points, so it was low (<5 points; 2 RCTs [18,28]) or medium (5–7 points; 3 RCTs [26,27,29]). Unfortunately, no RCTs with high methodological quality (8–10 points on the PEDro scale) were identified. The methodological quality of both pilot CTs (neither of which had a control group) assessed with the 16-point MINORS was low (6 points) [30,31]. Therefore, the conclusions regarding the influence of physical exercise on static and dynamic balance in women treated for BC presented by the reviewed trials’ authors were backed by weak or moderate scientific evidence. More high-quality clinical research is needed to reach firm conclusions about which types of physical exercises have the potential to improve the static and dynamic body balance in women treated for BC and how to apply them.

The trials varied in the methodology of exercises used. The authors of two RCTs opted for strength exercises [26,27], while the remaining three RCTs [18,28,29] implemented training programmes consisting of endurance and sensimotor exercises, fitness exercises with elements of soccer, and Pilates exercises, respectively. The patients in the pilot CTs [30,31] were administered various aerobic, resistance, balance, flexibility, and balance exercises.

The trials also differed in the total length of interventions and the frequency and duration of exercises. In only two RCTs [27,28], patients were treated for the same period of 12 weeks. The authors of one of them had patients perform soccer fitness exercises twice daily for 45–60 min and supervised all sessions [28]. The post-intervention measurements showed that the EG demonstrated statistically significantly better static balance than the CG (ADL only). In the other trial [28], in which the EG performed strength training and the CG performed flexibility training, neither after 6 nor 12 months were any statistically significant differences in the groups’ statistic balance, gait efficiency, and dynamic balance observed. In this trial, however, patients exercised for 60 min during three weekly sessions. The CG exercised unsupervised, while the EG performed supervised sessions twice a weekly, while exercising once weekly unsupervised at their homes.

In the other three RCTs [18,26,29], patients exercised for 4 [29], 4.5 [18], and 24 [26] months. Sessions lasted 30–60 min and were carried out two to three times a week [18,26,29] by women in the EGs, whereas the CGs only continued their ADL [18,26,29]. In one trial [18], participants’ static balance significantly improved after 4 months of endurance and sensimotor exercises performed twice weekly (reductions in the sway area measured with a force platform were observed). After another 4.5 months, both static and dynamic body balance improved [18]. The authors of another trial [29] reported better dynamic balance in patients after 4 months of Pilates exercises (60 min/day; 3 days/week). In both trials [18,29], patients performed supervised exercises in fitness centres. The third trial’s intervention [26] consisted in 24 months of strength training (30–45 min/day; 2 days/week), during which patients exercised unsupervised at their homes for the first 8 months (according to the instructions they received) and then performed supervised exercises in fitness centres for the remaining 16 months. In this trial, it was only after 24 months of intervention that an improvement in dynamic balance in the EG was observed; neither after 6 nor 12 months was the EG statistically significantly different in dynamics from the CG [26].

The results of the cited RCTs [18,26,27,28,29] imply that 4–12 months of endurance training, sensimotor exercises, soccer fitness exercises, and Pilates exercises may help improve static and dynamic body balance in women treated for BC. Their dynamic balance was also able to be increased by strength exercises, but in that case, 24 months of training was required. As these conclusions are based on evidence coming from few clinical trials of low and medium methodological quality, their validity should be confirmed by other, high-quality RCTs using relatively similar physical training programmes. With this approach, it would be possible to determine which of these improve body balance in women treated for BC the most effectively. Future RCTs should also compare the effectiveness of supervised and unsupervised exercises in BC survivors treated for balance disorders.

In both pilot CTs [30,31], patients exercised for 3 months. In one of them, exercise sessions of 90 min were performed under supervision twice weekly [30]. In the second trial [31], the frequency and length of exercise sessions varied; in the first 1.5 months, supervised exercises and educational activities were performed once weekly (a session lasted 150 min). In the next 1.5 months, exercise sessions were held three times a week; one session of 150 min involved aerobic training (walking) and was performed by unsupervised patients; the other two sessions took place in fitness centres and were supervised. Both pilot CTs used multimodal training comprising aerobic [30,31], resistance [30,31], balance [30,31], and flexibility exercises [30]; static body balance improvements were noted after 4 months of intervention [30,31]. The authors of the pilot CTs [30,31] also observed the statistically significant betterment of patients’ functional gait efficiency and dynamic balance, as well as functional reach and dynamic balance. In one of the trials [31], dynamic balance needed less time to improve (6 weeks) [31]. Summing up, despite the authors of the pilot CTs using somewhat different exercise protocols, particularly regarding the weekly frequency and length of sessions, multimodal training used in both CTs similarly improved static and dynamic body balance in women treated for BC. It needs to be noted, however, that neither of the trials used a control group, which is a limitation of their results. Therefore, the methodology of exercise adopted by their authors and their findings need to be further verified by RCTs.

It is worth noting that the authors of the reviewed trials assessed patients’ body balance using different methods. Dynamic balance in all trials was measured by functional tests, but depending on the trial short physical performance battery (sPPB) [27], the mini balance evaluation systems test (MiniBEST) [29], the functional reach test (FRT) [30], The Timed Up and Go Test (TUG) [30], The Tandem Walk Test (TW) [31], and the timed backward tandem walk (TBTW) [26] were used. As well as dynamic balance, each of the tests also assesses other motor functions, including gait efficiency (sPPB, Fullerton ABS, MiniBEST, TUG, TW, TBTW) and functional range of reaching (FRT). Five of the seven reviewed trials assessed patients’ static body balance [18,27,28,30,31]. the one-leg stance test (One-LegST) was used in three trials [27,30,31]. The authors of the other two trials chose the flamingo balance test (FlamingoBT) [28] and posturometry involving a force platform [18]. In one trial [18] the Fullerton advanced balance scale (Fullerton ABS) measuring both static and dynamic balance was also used.

To the review authors’ knowledge, only one of the above tests, the Fullerton ABS, was validated for the assessment of static and dynamic body balance in women treated for BC. Its validation was performed by Wampler et al. [12] during a trial involving 20 women after chemotherapy for BC (mean age 50.34 ± 9.34 years) and 20 healthy women (mean age 49.60 ± 9.08 years). They demonstrated that the Fullerton ABS had excellent test–retest reliability (interclass correlation coefficient ICC = 0.98) and excellent interrater reliability (ICC = 0.98).

The other tests were not validated for assessing body balance in women treated for BC, but most of them were validated for people affected by health problems similar to those faced by BC survivors, such as elderly people with balance disorders (TUG, TW, sPPB, MiniBEST, FRT), stroke patients (TUG), and people with neuropathies (TW). The TUG was originally created to assess dynamic body balance, gait efficiency, and fall risk in the elderly, but now it is widely used to assess body balance and gait efficiency of people affected by Parkinson’s disease, strokes, multiple sclerosis, hip fractures, and Alzheimer’s disease. Its intra-rater reliability and inter-rater reliability are high (ICC of 0.92–0.99) [33]. The sensitivity and specificity of the test are estimated at 87% [34]. The sPPB validation rates are also high. According to researchers, the sPPB provides an objective measure of balance, lower extremity strength, and functional capacity, and its inter-rater ICC for older people (>60 years) is 0.81–0.91 [35]. Freiberger at al. [36], based on a systematic review of clinical studies, concluded that the that the inter-rater ICC of the sPPB for balance and gait disorders in the elderly was 0.70–0.99.

The MiniBEST is another test enabling the quantitative assessment systems whose impairment affects postural control and leads to poor functional balance. The MiniBEST was validated with a group of people aged 50 to 80, which included patients with unilateral and bilateral vestibular loss, Parkinson disease, and peripheral neuropathy [37], and had excellent inter-reliability with ICC 0.79–0.95 for bio-mechanical constraints, stability limits/verticality, anticipatory postural adjustments, postural reactions, sensors orientation, and stability of gait ranging from 0.79 to 0.95 [38]. The FRT assesses dynamic balance and predicts fall risk in elderly and frail adults [39], as well as in other human populations, including patients with neuropathies [40]. It has the test–retest reliability with ICC of 0.89, and inter-rater agreement on reach measurement of 0.98 [39,41]. The sensitivity of the FRT is rated at 76%, accuracy at 46%, specificity at 34%, positive predictive value at 33%, and negative predictive value at 77% [42].

The TW and TBTW allow the assessment of functional dynamic balance. The TW test is an objective and useful tool for evaluating dynamic balance and gait disturbance in patients with degenerative cervical myelopathy [43] or peripheral neuropathy [44]. Receiver operating characteristic (ROC) analysis comparing the number of steps between the control group (23–77 years; mean age 49.6 ± 16.0) and the subjects with peripheral neuropathy (mean age 60 ± 12.4 years) showed that the reliability of the TW was moderate for eyes open (ROC 0.81, 95%CI 0.70–0.93) and high for eyes closed (ROC 0.88, 95%CI 0.81–0.96) [44]. Whether the TBTW has been validated for women with BC or patients with similar health problems (relating to perimenopausal or postmenopausal age, older age, neuropathies, etc.) could not be determined by the review authors. Nelson et al. [45] reported significant correlations between the weekly results of the TBTW (r = 0.94, *p* = 0.001) after studying postmenopausal women aged 50–70 years. The Pearson’s r coefficient between the results of the TUG and TBTW tests, calculated by Haque et al. [46] for subjects aged 71.62 ± 5.39 years, was 0.61 at *p*-value of 0.01.

The one-legST test of less than 30 s is predictive of poor balance and fall risk in elderly people [47]. The test has excellent inter-rater reliability (gamma coefficient = 0.99–1.00) and good predictive validity (RR = 11.6; 95%CI: 1.7–80) for falls in older adults [47,48]. Springer et al. [49] confirmed its excellent inter-rater reliability after applying it to assess the postural balance of people at different ages, including 60-year-olds and older; the ICC of 0.994 (95%CI 0.989–0.996) for eyes open and ICC of 0.998 (95%CI 0.996–0.999) for eyes closed were obtained. Hurvitz et al. [50] found the one-legST to be useful in the clinical setting as a tool for confirming or excluding the presence of peripheral neuropathy, which may lead to the development of balance disorders in women treated for BC. In the study they conducted, abnormal unipedal stance time (<45 s) identified peripheral neuropathy with a sensitivity of 83% and a specificity of 71%, whereas normal unipedal stance time had a negative predictive value of 90%. Abnormal unipedal stance time was associated with an increased risk of peripheral neuropathy on univariate analysis (odds ratio = 8.8, 95% confidence interval = 2.5–31) [50]. Considering the above, The one-legST can be seen as a proven and reliable method for assessing static balance in women treated for BC.

The validity and reliability of the flamingo BT have been evaluated for relatively young people, including children (aged 9.1 ± 1.8 years) [51,52], university students (men and women aged 19.5 ± 2.7 and 19.4 ± 2.7 years, respectively) [53], people with bipolar disorder (men and women aged 43.07 ± 13.0 and 43.97 ± 10.2 years, respectively) [54], and alcohol abusers at risk of cardiovascular disease (men and women aged 40.8 ± 13.8 and 41.9 ± 12.1, respectively) [55]. The reliability of the flamingo BT in the cited studies was relatively high (ICC > 0.70; 95% CI) [51,52,53,54,55]. The downside of the test is that it requires relatively good physical fitness of the patients who perform it, especially an adequate ability to stretch the quadriceps femoris [53], which may make its application problematic when the postural balance of elderly patients is to be assessed.

Posturometry measures the movement of the centre of foot pressure (COP) using a force platform to assess a person’s postural stability. Sampling frequency may vary depending on the device. The COP sway area (expressed in cm^2^) that Vollmers et al. [18] measured at a sampling frequency of 1000 Hz is determined in subjects standing on one leg (both legs are tested) and evaluated by calculating a 95% predication ellipsis, which is independent of the sampling frequency. Body sway may depend on many factors and that its greater and smaller amounts do not necessarily reflect poorer or better postural control, respectively [56,57,58,59,60]. Accordingly, the range of COP and sway area obtained with a force platform may be insufficient to assess a person’s static balance. Studies assessing the reliability of sway measures during different postural control tasks in various age populations show that velocity is the most reliable measure of postural sway [61,62]. This observation has been corroborated by systematic review of 32 studies by Ruhe et al. [63], which also found that sway velocity is a more reliable parameter than the range of sway area.

The assessment of the reviewed studies’ results is hindered by the aforementioned variety of tests used by their authors to assess body balance. Future research should, therefore, make a wider use of tests that have been validated for assessing body balance in women treated for BC, such as the Fullerton AB (static and dynamic body balance) and the one-legST (static balance). Other tests may also be useful provided that they have high validity for assessing the static and dynamic postural balance in people with symptoms similar to those suffered by women with BC, e.g., the elderly, people with neuropathies, or peri- and post-menopausal women.

### 4.1. Summary

Although the authors of most studies [18,26,28,29,30,31] reported improvements in the static [18,28,30,31] and dynamic [18,26,29,30,31] body balance of their patients post-intervention, the small number of RCTs and pilot CTs published so far and the variety of training programmes and body balance assessment methods used by their authors prevent the drawing of clear conclusions regarding which programmes could be the most effective in improving the static and dynamic postural balance in women treated for BC.

The results of our systematic review of clinical trials cannot be verified by comparing them with the findings of other reviews or meta-analyses on the effect of physical exercise on body balance in women treated for BC, because we have not been able to identify any publications on this topic.

### 4.2. Limitations

The main limitation of the review is that the scarcity of studies investigating the effect of physical exercises on body balance in women treated for BC prevent the conducting of a meta-analysis. Other limitations include: (1) omitting trial reports other than those published in the English language; (2) low or medium methodological quality of the accepted trials; (3) trials using different methods to assess patients’ static and dynamic balance; (4) trials using different exercise programmes; (5) substantially different durations of the interventions (1.5–24 months); (6) the use of supervised and unsupervised exercise sessions; (7) certain trial authors failing to state the details of their patients’ clinical condition, which may have influenced the course of the intervention (e.g., BC stage and the type of surgical treatment received by the patients); and (8) certain interventions started while the patients were still in chemotherapy, while others began at different time points after it had already ended.

### 4.3. Future Research

Given the limited reliability of the reviewed studies, there is a need for RCTs of the highest methodological quality using comparable exercise programmes and static and dynamic balance assessment methods.

## 5. Conclusions

The results of the reviewed RCTs and pilot CTs are promising and support the use of physical exercises as a means of improving static and dynamic balance in women treated for BC. Considering, however, that this conclusion is based on two pilot CTs and five RCTs of low or moderate methodological quality, it should be taken as tentative and requires further, high-quality clinical research in order to be confirmed.

Drawing unambiguous conclusions from the reviewed trials is hindered by the diversity of exercise programmes and methodologies used by their authors, as well as by different methods they deployed to assess body balance in women with BC post-intervention. There is, therefore, an obvious need for high-quality RCTs using comparable training programs and diagnostic methods with the highest degree of validation to assess the static and dynamic balance in women treated for BC.

## Figures and Tables

**Figure 1 ijerph-20-03722-f001:**
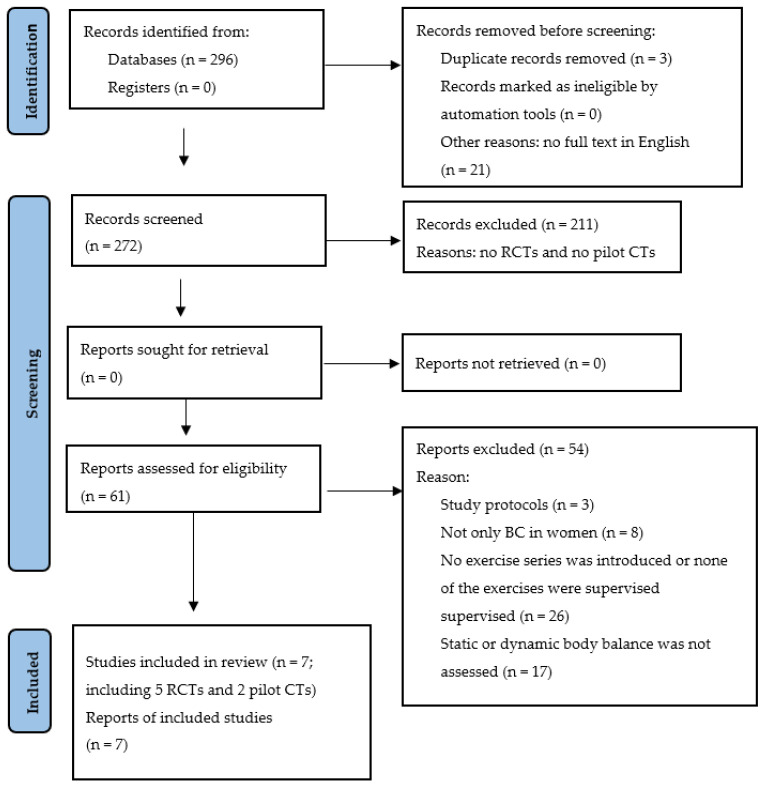
The literature search flow diagram (BC—breast cancer; RCTs—randomized clinical trials; pilot CTs—pilot clinical trials).

**Table 1 ijerph-20-03722-t001:** Summary of search strategies in both research databases used (PubMed and EBSCO).

Database	Search Strategy
PubMed EBSCO	((“breast cancer”) AND (“physical activity”) AND (“balance”)) OR ((“breast cancer”) AND (“physical activity”) AND (“postural balance”)) OR ((“breast cancer”) AND (“physical activity”) AND (“fall risk”)) OR ((“breast cancer”) AND (“physical exercise”) AND (“balance”)) OR ((“breast cancer”) AND (“physical exercise”) AND (“postural balance”)) OR ((“breast cancer”) AND (“physical exercise”) AND (“fall risk”)) OR ((“breast cancer”) AND (“exercise training”) AND (“balance”)) OR ((“breast cancer”) AND (“exercise training”) AND (“postural balance”)) OR ((“breast cancer”) AND (“exercise training”) AND (“fall risk”))

**Table 2 ijerph-20-03722-t002:** The results of methodological quality assessment of RCTs based on the PEDro scale.

Authors	PEDro Scale Criteria (1 = Conditions Met, 0 = Criteria Not Fulfilled)										Total
	*1*	*2*	*3*	*4*	*5*	*6*	*7*	*8*	*9*	*10*	
Twiss et al. (2009) [26]	1	0	1	0	0	0	1	1	1	0	5
Winters-Stone et al. (2012) [27]	1	1	1	0	0	1	0	1	1	1	7
Vollmers et al. (2018) [18]	1	0	0	0	0	0	0	0	1	1	3
Uth et al. (2021) [28]	1	0	0	0	0	0	0	1	1	1	4
de Bem Fretta et al. (2021) [29]	1	0	1	0	0	0	0	1	1	1	5

**Table 3 ijerph-20-03722-t003:** The results of methodological quality assessment of pilot CTs based on the MINORS.

MINORS Criteria	Points (0—Not Reported; 1—Reported but Inadequate; 2—Reported and Adequate)	
	Lee et al. 2016 [30]	Foley et al. 2016 [31]
1. A clearly stated aim	2	2
2. Inclusion of consecutive patients	1	2
3. Prospective collection of data	2	1
4. Endpoints appropriate to the aim of the study	1	1
5. Unbiased assessment of the study endpoint	0	0
6. Follow-up period appropriate to the aim of the study	0	0
7. Loss to follow up less than 5%	0	0
8. Prospective calculation of the study size	0	0
Additional criteria in the case of comparative studies	no control group	no control group
9. An adequate control group	0	0
10. Contemporary groups	0	0
11. Baseline equivalence of groups	0	0
12. Adequate statistical analyses	0	0
In total	6	6

The global ideal score being 16 for non-comparative studies and 24 for comparative studies. MINORS—Methodological Index For Non-Randomized Studies.

**Table 4 ijerph-20-03722-t004:** Characteristics of the groups in randomized clinical trials.

Article	No. of Women	Age (Years) Mean ± SD	Stage of Breast Cancer (Number; %)					Surgery (Number; %)			Chemo-Therapy	Radio-Therapy	Immuno-Therapy	Hormone Therapy
			0	I	II	III	IV	Total	Conserving	Mastectomy	[Number; %]			
* Twiss et al. (2009) [26] 5 ***	223 EG = 110; CG = 113	Range: 41–75 T: 58.69 ± 7.5 (no data for EG and CG)	Yes	Yes	Yes	No	No	98.2%	n/d	n/d	T: 67.7%	T: 45.3%	n/d	T: 39.5%
** Winters-Stone et al. (2012) [27] 7 ***	106 EG = 52 CG = 54	Range: 53–83 62.3 ± 6.7 62.2 ± 6.7	7.7% 3.7%	38.5% 40.7%	48.1% 35.2%	1.9% 9.3%	0% 0%	n/d	n/d	n/d	T: 60.4% 61.5% 59.3%	T: 82.8% 92.3% 83.3%	T: 15.1% 17.3% 13% a	T: 41.5% 42.3% 40.7%
Vollmers et al. (2018) [18] 3 ***	36 EG = 17 CG = 19	Range: 20–68 48.56 ± 11.94 52.39 ± 10.14	n/d	n/d	n/d	n/d	n/d	n/d	n/d	n/d	100%	n/d	n/d	n/d
Uth et al. (2021) [28] 4 ***	68 EG = 46 CG = 22	Range: n/d 47.4 ± 9.4 50.0 ± 9.3	0% 0%	39.1% 45.5%	39.1% 54.5%	21.7% 0%	0% 0%	100%	58.7% 59.1%	41.3% 40.2%	T: 89.7% 93.5% 81.8%	T: 80.9% 82.6% 77.3%	T: 30.9% 32.6% 27.3%	T: 79.4% 78.3% 81.8%
de Bem Fretta et al. (2021) [29] 5 ***	34 EG = 18 CG = 16	Range: n/d 53.33 ± 8.58 57.5 ± 13.02	n/d	n/d	n/d	n/d	n/d	100%	50% 68.7%	50% 31.3%	T: 91.2% 88.9% 93.7%	T: 70.6% 72.2% 68.7%	n/d	n/d

EG—experimental group; CG—control group; T—Total; SD—standard deviation; n/d—no data. * The Twiss et al. study involved women treated for stage 0–II BC, but the specific numbers of patients with particular BC stages were not reported; ** In the Winters-Stone et al. study, BC stage was not reported for 3.8% of the patients in the EG and 11.1% in the CG; *** PEDro score.

**Table 5 ijerph-20-03722-t005:** Characteristics of the groups in pilot clinical trials.

	Number of Women	Age (Years) Mean ± SD	Stage of Breast Cancer (Number; %)					Surgery (Number; %)			Chemo-Therapy	Radio-Therapy	Immuno-Therapy	Hormone Therapy
			0	I	II	III	IV	Total	Conserving	Mastectomy	[%]			
Foley et al. (2016) [30] * 6	52	59.7 ± 10.4 Range: 46–82	n/d	n/d	n/d	n/d	n/d	52; 100%	n/d	n/d	44 85%	39 75%	n/d	20 39.5%
Lee et al. (2016) [31] * 6	56	53.8 ± 9.6 Range: 34–73	0 0%	14 25%	31 55.4%	11 19.6%	0 0%	n/d	n/d	n/d	Both chemo- and radiotherapy: 38; 67.9% Chemo- or radiotherapy only: 18; 32.1%		n/d	n/d

SD—standard deviation; n/d—no data. * MINORS score.

**Table 6 ijerph-20-03722-t006:** Body balance assessment methods in randomized clinical trials and pilot clinical trials.

	Methods of Body Balance Assessment
Twiss et al. (2009) [26] RCT	Dynamic balance and gait efficiency: the timed backward tandem walk
Winters-Stone et al. (2012) [27] RCT	Static balance: the one-leg stance test Dynamic balance and gait efficiency: short physical performance battery
Vollmers et al. (2018) [18] RCT	Static and dynamic balance: the Fullerton advanced balance scale Static balance: the sway area (cm^2^) in monopedal stance for left and right leg with use of force platform
Uth et al. (2021) [28] RCT	Static balance: the flamingo balance test
de Bem Fretta et al. (2021) [29] RCT	Dynamic balance and gait efficiency: the mini balance evaluation system test
Foley et al. (2016) [30] Pilot CT	Static balance: the one-leg stance test Functional reach and dynamic balance: the functional reach test Dynamic balance and gait efficiency: the timed get up and go test
Lee et al. (2016) [31] Pilot CT	Static balance: the one-leg stance test Dynamic balance and gait efficiency: the tandem walk test

RCT—randomized clinical trial; pilot CT—pilot clinical trial.

**Table 7 ijerph-20-03722-t007:** Methods of randomized clinical trials and their results regarding the static and dynamic balance of the body.

	Intervention	Physical Training Methodology in the EG		Measurement Time	Measures	Results
		Frequency	Duration			
Twiss et al. (2009) [26] 5 *	EG: strength training CG: only ADL	Strenght training was performed twice weekly for 30–45 min per day. Unsupervised, home-based exercises were performed in the first 8 months followed by supervised exercises at a fitness centre for the remaining 16 months.	24 months	Baseline, months 6, 12, and 24 Follow up: month 12	Dynamic balance ^1^	At month 24, the dynamic balance was statistically significantly greater in the EG than in the CG (*p* = 0.010). At months 6 and 12, no statistically significant differences (*p* > 0.05) between groups were noted.
Winters-Stone et al. (2012) [27] 7 *	EG: strength training CG: only flexibility training	Strenght training was performed three times weekly for 60 min per day. For 2 days, supervised exercises were performed at a training centre and for 1 day unsupervised, home-based exercises were performed.	12 months	Baseline, months 6 and 12	Static balance ^2^ Dynamic balance ^3^	The static balance ^2^, the gait efficiency, and dynamic balance ^3^ did not differ statistically significantly between the groups either at 6 or 12 months (*p* > 0.05).
Vollmers et al. (2018) [18] 3 *	EG: endurance and sensimotor exercises CG: only ADL and recommendations for individual physical activity	Endurance and sensimotor exercises were performed twice weekly. All exercises were supervised and were performed at a training centre. The duration of exercising dependent on the day’s the training programme.	18 weeks (4.5 months)	Baseline, week 12 (end of chemotherapy), week 18 (6 weeks post-chemotherapy)	Static balance ^4^ Static and dynamic balance ^5^	At week 12 (the end of chemotherapy) the static balance in the EG was significantly greater than in the CG (*p* = 0.003 for sway area in stance on the left leg and *p* = 0.01 for sway area in stance on the right leg). At week 18 (6 weeks post-chemotherapy), both static and dynamic balance were significantly greater in the EG than in the CG (*p* = 0.004).
Uth et al. (2021) [28] 4 *	EG: soccer fitness exercises CG: only ADL	Soccer fitness exercises were performed twice weekly for 45–60 min per day. All exercises were supervised and were performed at a training centre.	12 months	Baseline, months 6 and 12	Static balance ^6^	At month 6, no statistically significant difference (*p* > 0.05) between groups was noted. At month 12, the static balance in the EG was statistically significantly greater than in the CG (*p* = 0.021).
de Bem Fretta et al. (2021) [29] 5 *	EG. Pilates training CG: only ADL	Piltates training was performed for 3 days a week, for 60 min per day. All exercises were supervised and were performed at a training centre.	4 months	Baseline, week 16	Dynamic balance ^7^	At week 16, the dynamic balance ^7^ was statistically significantly greater in the EG than in the CG (*p* = 0.034).

EG: experimental group; CG—control group; ADL—activity of daily living; * PEDro score; ^1^ the timed backward tandem walk; ^2^ the one-leg stance test; ^3^ the short physical performance battery; ^4^ posturometry—sway area (cm^2^) in monopedal stance on the left and right legs; ^5^ the Fullerton advanced balance scale; ^6^ the flamingo balance test; ^7^ the mini balance evaluation systems.

**Table 8 ijerph-20-03722-t008:** Methods of pilot clinical trials and their results regarding the static and dynamic balance of the body.

Author	Intervention	Physical Training Methodology		Measurement Time	Measures	Results
		Frequency	Duration			
Foley et al. (2016) [30] * 6	Multimodal exercise program (aerobic, resistance, balance, and flexibility exercises)	Training was performed twice weekly for 90 min per day. All exercises were supervised and were performed at a training centre.	3 months	Baseline, week 12	Static balance ^1^ Functional reach and dynamic balance ^2^ Dynamic balance ^3^	Static balance improved (*p* < 0.001) at week 12. Functional reach and dynamic balance improved (*p* < 0.001) at week 12. Functional dynamic balance improved (*p* < 0.001) at week 12.
Lee et al. (2016) [31] * 6	Multimodal exercise program (aerobic, resistance and balance exercises)	For the first 1.5 months, the educational classes and supervised physical exercises were performed once a week. The total time of educational classes and exercises was 150 min per day. For the next 1.5 months, the unsupervised home-based exercises were performed: - once weekly for 150 min (aerobic training) - twice or three times weekly (resistance training; duration depended on the number of exercises performed) - twice or three times weekly (balance training; duration depended on the number of exercises performed)	1.5 months 1.5 months	Baseline, week 6 (1.5 months), week 12 (3 months)	Static balance ^1^ Dynamic balance ^4^	Static balance improved (*p* < 0.05) at week 12. Dynamic balance improved (*p* < 0.01) at week 6 and 12.

^1^ the one-leg stance test; ^2^ the functional reach test; ^3^ the timed up and go test; ^4^ the tandem walk test; * MINORS score.

## Data Availability

The data presented in this study can be obtained by contacting the corresponding author.

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
