# Peer review of "Effect of Physical Activity on Static and Dynamic Postural Balance in Women Treated for Breast Cancer: A Systematic Review"

_ijerph, 2023, doi:10.3390/ijerph20043722_

Round 1

Reviewer 1 Report

The manuscript entitled "Effect of Physical Activity on Static and Dynamic Postural Balance in Women Treated for Breast Cancer. A Systematic Review of Clinical Trials" studies a very interesting topic concerning women's health. Breast cancer is one of the most common cancers in women and its effects after treatment are quite physically conditioning for survivors.

Having reviewed the manuscript, I would like to make the following observations.

The title does not seem to me to be totally appropriate, since it is not mentioned that pilot studies are also included. With the title presented, it could lead to confusion; "Clinical Trials" should be eliminated.

In the reading of the manuscript there are acronyms that when they are introduced for the first time their meaning is not explained, although it is known, it should be explained.

The objective described in the methodology is not reflected in the abstract.

As for the methodology, the title states that it is a systematic review, but it is not stated that it follows the PRISMA guidelines, which all systematic reviews must follow.

The publication of the review protocol, the inclusion and exclusion criteria, etc., is not included. 

In point 2.1. it states the following sentence "... and the online resources of gay publications using ...". My knowledge of English may not be sufficient, I would be grateful if you could explain the meaning of this sentence.

Can you include a table with the search strategy for each of the databases studied? This table should be provided as supplementary material.

Studying only English-language articles is an important bias that you should point out as a limitation of your manuscript.

In the results, perform the search according to the inclusion and exclusion criteria, the flow chart will be clearer. You have performed the search without applying the inclusion and exclusion criteria, hence the 7-digit numbers.

By applying the criterion of only texts in English, 1,364,160 documents in other languages that could have contributed a lot of information to this review are left out.

The flow chart should be revised and adapted to PRISMA standards.

Was the inclusion criterion that the methodological quality of the selected articles was not important? In my opinion, articles with low methodological quality should not have been selected. In any case, it should be noted in limitations. The methodological quality of the pilot studies should be evaluated based on validated scales that assess the methodological quality of a project.

Section 3.4. with the explanation of each of the selected articles is too long, it should be summarized.

The section "Conclusion" should be called "Discussion".

Author Response

We kindly thank you for reviewing our article and for all the reviewers' comments that contributed to improving the quality of our manuscript. We hope that the changes we made to the text of the article will be appreciated by the reviewers and editors.

Considering comments from reviewers, we have made extensive changes to the article to make it more readable and in line with PRISMA guidelines, namely:

  • We have changed the abstract so that its structure is in line with the PRISMA guidelines;
  • In the Methods chapter, we have added the following subsections: 1. Protocol and Ethics proclamations; 2.5 Data items, and 2.7 Synthesis methods;
  • In the Results chapter, we have added the following subsections: 3. Group characteristics and 3.4 Measures;
  • We rewrote the discussion and added the "limitations" and "future research" subsections to the Discussion chapter;
  • In accordance with its purpose, we focused the review results on the static and dynamic balance of the body, removing extensive information on other variables assessed in the studies included in the review, thanks to which the article became clearer;
  • We assessed the methodological quality of the pilot CTs included in the review using The Methodological Index for Non-Randomized Studies (MINORS);
  • We added Table 1 entitled: “Summary of search strategies in different databases used” and corrected the other tables to make them more readable;
  • We adjusted the literature search flow diagram (Figure 1.) to comply with the PRISMA guidelines. We have added information about registration of the review in PROSPERO.

Detailed responses to reviewers' comments

Review 1

  1. The title does not seem to me to be totally appropriate, since it is not mentioned that pilot studies are also included. With the title presented, it could lead to confusion; "Clinical Trials" should be eliminated.

Answer: The title has been changed.

  1. In the reading of the manuscript there are acronyms that when they are introduced for the first time their meaning is not explained, although it is known, it should be explained.

Answer: We've explained all the acronyms the first time you use them.

  1. The objective described in the methodology is not reflected in the abstract.

Answer: The abstract has been changed in accordance with PRISMA guidelines.

  1. As for the methodology, the title states that it is a systematic review, but it is not stated that it follows the PRISMA guidelines, which all systematic reviews must follow.

Answer: Information has been added in the abstract and in the Methods chapter in subsection: 2.1. Protocol and Ethics proclamations.

  1. The publication of the review protocol, the inclusion and exclusion criteria, etc., is not included. 

Answer: The study protocol was not published separately. But the study was registered in PROSPERO. The information about the registration of the study was added in the Methods chapter, in the subchapter: 2.1. Protocol and Ethics proclamations

  1. In point 2.1. it states the following sentence "... and the online resources of gay publications using ...". My knowledge of English may not be sufficient, I would be grateful if you could explain the meaning of this sentence.

Answer: Unfortunately, the knowledge of the English language of the authors of the manuscript is not the best, but at this point we certainly meant articles from the so-called "grey zone". The error was corrected in the manuscript, in the subchapter: 2.3 Search strategy and selection process (Methods chapter).

  1. Can you include a table with the search strategy for each of the databases studied? This table should be provided as supplementary material.

Answer: The table was added in the Methods chapter and is titled: “Table 1. Summary of search strategies in different databases used”

  1. Studying only English-language articles is an important bias that you should point out as a limitation of your manuscript. By applying the criterion of only texts in English, 1,364,160 documents in other languages that could have contributed a lot of information to this review are left out.

Answer: The limitation is listed in the Limitations section.

  1. In the results, perform the search according to the inclusion and exclusion criteria, the flow chart will be clearer. You have performed the search without applying the inclusion and exclusion criteria, hence the 7-digit numbers. The flow chart should be revised and adapted to PRISMA standards.

Answer: The description of the database search method and the flow chart have been changed in accordance with the PRISMA guidelines.

  1. Was the inclusion criterion that the methodological quality of the selected articles was not important? In my opinion, articles with low methodological quality should not have been selected. In any case, it should be noted in limitations. The methodological quality of the pilot studies should be evaluated based on validated scales that assess the methodological quality of a project.

Answer: We did not include the methodological quality of the RCTs and the pilot CT as the criteria for inclusion in the review because we did not want to unduly limit the number of studies included in the review - as it turns out, few studies on the impact of physical activity on static and dynamic body balance in women treated for BC has been published so far.

We emphasized the low methodological quality of the studies included in the review in the text of the article (among others in the discussion and conclusions). And additionally, we indicated this fact in the Limitations section.

We completed the assessment of the methodological quality of the pilot studies included in the review. This assessment was performed using The Methodological Index for Non-Randomized Studies (MINORS). The results of the assessment are described in the Results chapter and presented in Table 3.

  1. Section 3.4. with the explanation of each of the selected articles is too long, it should be summarized.

Answer: Descriptions of individual trials in sections 3.6 and 3.7 have been shortened.

  1. The section "Conclusion" should be called "Discussion".

Answer: A Discussion section has been added to the manuscript.

Reviewer 2 Report

·         Affiliation of author #3. Did you mean to incorporate a quote here?

·         Pg 2, ln 54: “The downside of BC therapies…” consider using a different word other than downside, ie: effect, negative effects. “Downside” seems awkward.

·         Pg 2: ln 62 – how is CVD related to balance? That connection isn’t very clear.

·         I like that you didn’t omit pilot CT’s as this area of research is sparse as it is.

·         Pg 3, ln 99 – for method assessment – could you please provide more detail here? Were you looking for particular tests used? Or were you examining the research design itself? More information in this regard would be helpful, especially because balance and posture can be measured with various tests. Comparing results of studies that used different posture/balance assessment can be difficult, making for difficult interpretation.

·         Pg 5 ln 179 – for this study which consisted of at home, unsupervised exercise, did the authors report on adherence to the program?

o   This study used timed backward tandem walk test to measure dynamic balance. Posture was not measured?

·         Pg 7, ln 269 – “smaller sway in monopodal stance…” isn’t this a measure of static balance and monopodal = single leg stance? This part is confusing.

·         Pg  7, ln278 Uth et al. used soccer training elements. Above when you first introduce this paper, you called it “football” (ln 142). Consider listing soccer in parentheses (ln 142) as it took me a minute to realize you were referring to soccer and not American Football. Whichever way you go, it’d be a good idea to be consistent.

·         The description of the intervention each study used was excellent.  

·         I’m having a difficult time matching up your findings to your aims. Your first aim is to determine physical benefit of physical exercise for static and dynamic balance. This review is difficult since the papers reviewed used different types of exercise, which means for the sake of the review, it’s difficult to ascertain the benefits.

·         Your second practical goal of determine exercise methods that would best benefit women’s body balance/reduce risk of falls – I’m not sure the review was able to do this. The different balance/postural tests were so different among studies and the variable results makes it difficult to draw overarching conclusions.

·         While the methods seem appropriate, the resultant articles are difficult to compare.

·         The conclusion seems to repeat the results. What are the conclusions of the study?

o   Perhaps a better approach would be to highlight the difficulty of comparison among studies due to the different assessment used. For example, it’s nearly impossible to compare findings of the study that recorded the # of falls to a study that use a force platform. This itself is not the fault of the authors, but is a limitation of the types of research that have been done. This should be mentioned in the paper.

o   It would be helpful to say which exercise intervention was most promising.

o   Can authors make a recommendation of which measurement should be used to assess posture? Balance?

o   Can authors add additional research showing which measurements correlate to outcome/practical measures?

·         In the methods, more description of the tests used and what it consists of would be helpful. It would be helpful for authors to compare similar tests to each other. This would help the reader to follow along.

·         The mere fact that the papers described used such varied methods makes this review paper difficult to write. Much more detail is needed to help the reader understand similarities among the papers, which could help the author to draw large, overarching conclusions. As it stands, the paper is difficult to follow.

Author Response

We kindly thank you for reviewing our article and for all the reviewers' comments that contributed to improving the quality of our manuscript. We hope that the changes we made to the text of the article will be appreciated by the reviewers and editors.

Considering comments from reviewers, we have made extensive changes to the article to make it more readable and in line with PRISMA guidelines, namely:

  • We have changed the abstract so that its structure is in line with the PRISMA guidelines;
  • In the Methods chapter, we have added the following subsections: 1. Protocol and Ethics proclamations; 2.5 Data items, and 2.7 Synthesis methods;
  • In the Results chapter, we have added the following subsections: 3. Group characteristics and 3.4 Measures;
  • We rewrote the discussion and added the "limitations" and "future research" subsections to the Discussion chapter;
  • In accordance with its purpose, we focused the review results on the static and dynamic balance of the body, removing extensive information on other variables assessed in the studies included in the review, thanks to which the article became clearer;
  • We assessed the methodological quality of the pilot CTs included in the review using The Methodological Index for Non-Randomized Studies (MINORS);
  • We added Table 1 entitled: “Summary of search strategies in different databases used” and corrected the other tables to make them more readable;
  • We adjusted the literature search flow diagram (Figure 1.) to comply with the PRISMA guidelines. We have added information about registration of the review in PROSPERO.

Detailed responses to reviewers' comments

Review 2

  1. Affiliation of author #3. Did you mean to incorporate a quote here

Answer: It is not a quote, but the name of a student science association, so the quotation marks were redundant and the error has been corrected.

  1. Pg 2, ln 54: “The downside of BC therapies…” consider using a different word other than downside, ie: effect, negative effects. “Downside” seems awkward.

Answer: The error has been corrected and now the sentence reads: “Breast cancer therapies frequently lead to complications that hinder the recovery of patients and affect their quality of life.”

  1. Pg 2: ln 62 – how is CVD related to balance? That connection isn’t very clear. ?????

Answer: Indeed, this information is redundant and has been removed from the manuscript.

  1. I like that you didn’t omit pilot CT’s as this area of research is sparse as it is. – tkank you

Answer: Thank you for your comment.

  1. Pg 3, ln 99 – for method assessment – could you please provide more detail here? Were you looking for particular tests used? Or were you examining the research design itself? More information in this regard would be helpful, especially because balance and posture can be measured with various tests. Comparing results of studies that used different posture/balance assessment can be difficult, making for difficult interpretation.

Answer: We did not consider methods of assessing body balance as an inclusion criterion in the review, therefore we included studies in which different methods of assessing body balance were used in the review. We are aware that the conclusions of the review would be more reliable if they were based on studies that used a uniform assessment of body balance. However, due to the small number of studies published so far, it was unfortunately not possible to apply such a criterion.

In order to draw attention to the variety of body balance assessment methods, we added a subchapter: 3.4 Measures and tables 6 entitled: "Body balance assessment methods in randomized clinical trials and pilot clinical trials" in the "Results" chapter.

The variety of methods for assessing static and dynamic balance was emphasized in the "Limitations" section and in the "Future research" section we indicated the need for further research in which there will be unified methods for assessing body balance.

  1. Pg 5 ln 179 – for this study which consisted of at home, unsupervised exercise, did the authors report on adherence to the program? This study used timed backward tandem walk test to measure dynamic balance. Posture was not measured?

In a study by Twiss et al. [26], the authors provided information on participation in the exercises. We added this information to the description of the study by Twis et al. [26] in section: 3.6. and it reads as follows: "Adherence to exercises was measured using self-report of number of prescribed sessions attended and participants’ reports of falls. Mean adherence over 24 months was 69.4%. Mean 24-month adherence to exercises was 69.4%. Fifty of 110 women attended exercise sessions more than 80% of the time. Mean percentage adherence for 8 months of homebased exercises was 79.7% and for 16 months of fitness center exercises, 60.6%.”

In a study by Twiss et al. The Timed Backward Tandem Walk was only used to assess dynamic body balance (quote from the article by Twiss et al.: "Dynamic Balance was assessed using the Timed Backward Tandem Walk. Participants placed one foot behind the other and walked backward as fast as possible over a 20-foot course"). As the result of the test, the authors assumed the time to walk backwards for a distance of 20 feet, measured in seconds.

  1. Pg 7, ln 269 – “smaller sway in monopodal stance…” isn’t this a measure of static balance and monopodal = single leg stance? This part is confusing.

Answer: In a study by Vollmers et al. [18] the static balance of the body was assessed using posturometry. Sway are (cm2) were assessed while standing on one leg, separately for the left and right legs. An appropriate explanation was added in the text to the description of the study by Vollmers et al. [18] in section 3.6, and in subsection 3.4 "measures" and in table 6.

  1. Pg 7, ln278 Uth et al. used soccer training elements. Above when you first introduce this paper, you called it “football” (ln 142). Consider listing soccer in parentheses (ln 142) as it took me a minute to realize you were referring to soccer and not American Football. Whichever way you go, it’d be a good idea to be consistent.

Answer: The description of the exercises has been standardized to the name soccer.

  1. The description of the intervention each study used was excellent.

Answer: Thank you for your comment.

  1. I’m having a difficult time matching up your findings to your aims. Your first aim is to determine physical benefit of physical exercise for static and dynamic balance. This review is difficult since the papers reviewed used different types of exercise, which means for the sake of the review, it’s difficult to ascertain the benefits.

Answer: The main objective of the review was to find out if and what kind of exercise improves body balance in women treated for BC. And of course, according to the reviewer's comment, we should concentrate on this topic. It was unnecessary for us to describe the effects of exercise on other variables besides body balance. The manuscript has been revised by us, redundant information has been removed (both from the text and tables) and we hope that now the article is more readable, factual and understandable.

The variety of exercises used in the research is highlighted in the Discussion section. It was also indicated that due to the small number of RCTs and pilot CTs conducted so far, and due to the large variety of exercises used in the studies, it is not yet possible to indicate which exercises improve the body balance in women treated for BC to the greatest extent.

  1. Your second practical goal of determine exercise methods that would best benefit women’s body balance/reduce risk of falls – I’m not sure the review was able to do this. The different balance/postural tests were so different among studies and the variable results makes it difficult to draw overarching conclusions.

Answer: We fully agree with the reviewer's opinion. Unfortunately, the practical goal that we indicated in the previous version of the manuscript is not achievable, and therefore this goal was removed from the review. And the corresponding comment on exercise variety as mentioned above has been added to the discussion.

  1. While the methods seem appropriate, the resultant articles are difficult to compare. The conclusion seems to repeat the results. What are the conclusions of the study?

o   Perhaps a better approach would be to highlight the difficulty of comparison among studies due to the different assessment used. For example, it’s nearly impossible to compare findings of the study that recorded the # of falls to a study that use a force platform. This itself is not the fault of the authors, but is a limitation of the types of research that have been done. This should be mentioned in the paper.

o   It would be helpful to say which exercise intervention was most promising.

Answer: The discussion has been adjusted according to the reviewer's comments. Subsections Limitations and Future research have also been added.

  1. Can authors make a recommendation of which measurement should be used to assess posture? Balance? Can authors add additional research showing which measurements correlate to outcome/practical measures? In the methods, more description of the tests used and what it consists of would be helpful. It would be helpful for authors to compare similar tests to each other. This would help the reader to follow along.

Answer: Unfortunately, we are not competent to make scientific pronouncements on the best methods for assessing body balance, and our review also does not authorize us to draw conclusions on this subject. Nevertheless, the reviewer's comment inspired us to review studies on body balance in the future or to search for such reviews in scientific databases. In response to a reviewer's comment: 1) In the Methods chapter, we added a subchapter: Measures and Table 6, in which we listed the body balance assessment methods used in the research; 2) In the Discussion chapter, we included a commentary on the variety of body balance assessment methods used in the studies and indicated the need to standardize this diagnostics in further clinical trials.     

  1. The mere fact that the papers described used such varied methods makes this review paper difficult to write. Much more detail is needed to help the reader understand similarities among the papers, which could help the author to draw large, overarching conclusions. As it stands, the paper is difficult to follow.

Answer: We've sorted out the manuscript. We removed unnecessary information from the manuscript regarding the assessment of other variables besides body balance. We have introduced separate subchapters in which, among others, we have indicated the main issues discussed in the review (2.5 Data items), described the group characteristics (3.3. Group characteristics), characterized the methods of body balance diagnostics (3.4 Measures), characterized the methods of physical exercise used in the research (3.5 Methods of physical activity). We added a Discussion chapter, in which we tried to summarize, among other things, the methodology of exercises and the methodology of body balance assessment. We also identified the limitations of the review and guidelines for further research. However, if the changes introduced by us, in the opinion of the reviewer, are insufficient to make the review more understandable, we kindly ask for further comments that will allow us to improve the scientific and practical quality of the review.

Round 2

Reviewer 1 Report

The authors have carried out all the recommendations I made. The manuscript is now PRISMA compliant.

I consider it valid for publication.

Author Response

Review 1

I consider it valid for publication.

Answer: Thank you for your positive comment on our manuscript

Reviewer 2 Report

Abstract is much improved

Ln 52 – consider removing, “cause…female BC.” This sentence as it’s written is a bit awkward.

Ln 57 – consider using ‘prescribed’ instead of “required.” Some BrC patients may not think of them as required as they have the option to turn down treatment.

Ln 65 – missing C in complications?

Ln 63 – aim is clear and can be accomplished. Great edit.

Ln 112 – in the PDF version I have, it looks like the table is replicated twice, with same content repeated in the 2nd box?

Figure 1 – in my version, the “n” in identification in blue box is cut off.

Table 1- consider adding 1= condition met, = no/criteria not fulfilled?

Table 4 – what does n/d stand for? Please define.

Table 6- thank you for adding this table. This adds much needed clarity

Section 3.5 also add much clarity

Ln 317 – add ‘years’ after mean age.

Tables 7 and 8  - are difficult to read – could you please widen cells with text?

Ln 412- please undo the superscript text

Discussion – I think it would be better to state overall conclusion here…PA was effective in most but not all studies with limitations being… in other words, summarize the results. Ie: taken together, results of the systematic review suggest that….

I would not go too deep into IGFBP’s and CRP (biology) as its pretty far from balance / postural data.

Starting with ln 436- this is where you hone in on interpreting the results. Consider starting this discussion at this point.

Ln 434 – there’s a misplaced z in this sentence.

Ln 467- excellent comparison here. Can you make more comparisons among the other studies? Ln 474 – you do a great job of comparisons.

Revisit paragraph structure in the discussion. There are usually 3 sentences minimum in each paragraph. You have the components here, but the discussion can be reformatted/ rearranged to allow for better flow.

Ln 500 – can you spell out what high quality would be? Ample statistical power? Which balance test would you suggest to use, based on what you reviewed?

Conclusion  - great conclusion statement.

Author Response

Review 2

We kindly thank you for your comments. In response to the reviewer's comments below, all new text changes have been marked in blue.

  1. Abstract is much improved

Answer: Thank you for your positive comment.

  1. Ln 52 – consider removing, “cause…female BC.” This sentence as it’s written is a bit awkward.

Answer: Sentence: “New and enhanced therapies against BC, introduced with advances in medicine and the rising awareness of the importance of cancer screening cause that despite the increasing incidence of female BC, more and more women are treated early enough to have a good chance of recovery” has been changed to: "New and enhanced therapies against BC, and the rising awareness of the importance of cancer screening cause that more and more women with BC are treated early enough to have a good chance of recovery."

  1. Ln 57 – consider using ‘prescribed’ instead of “required.” Some BrC patients may not think of them as required as they have the option to turn down treatment.

Answer: The sentence has been corrected according to the reviewer's comment.

  1. Ln 65 – missing C in complications?

Answer: The error has been corrected.

  1. Ln 63 – aim is clear and can be accomplished. Great edit.

Answer: Thank you for your positive comment.

  1. Ln 112 – in the PDF version I have, it looks like the table is replicated twice, with same content repeated in the 2nd box?

Answer: In both research databases (PubMed and EBSCO) the keyword search strategy was the same, hence this repetition appeared in Table 1 in the PubMed and EBSCO boxes. But indeed this repetition is redundant (and may confuse the reader) and therefore Table 1 has been corrected accordingly.

  1. Figure 1 – in my version, the “n” in identification in blue box is cut off.

Answer: The error in Figure 1 has been corrected.

  1. Table 1- consider adding 1= condition met, = no/criteria not fulfilled?

Answer: We guess that the reviewer's comment concerns Table 2; the text in table 2 has been corrected according to the reviewer's comment.

  1. Table 4 – what does n/d stand for? Please define.

Answer: In Tables 4 and 5, the abbreviation "n/a" is defined as "no data".Appropriate explanations are placed under Tables 4 and 5.

  1. Table 6- thank you for adding this table. This adds much needed clarity

Answer: Thank you for your comment.

  1. Section 3.5 also add much clarity

Answer: Thank you for your comment.

  1. Ln 317 – add ‘years’ after mean age.

Answer: The missing word "years" was added on line 317. Similarly, "years" was added elsewhere in the text where it was missing.

  1. Tables 7 and 8 - are difficult to read – could you please widen cells with text?

Answer: Tables 7 and 8 have been corrected according to the reviewer's comments.

  1. Ln 412- please undo the superscript texth

Answer: The superscript has been removed.

  1. Great conclusion statement.

Answer: Thank you for your comment.

All of the reviewer's comments on the discussion below have been included. The discussion has been rewritten. Its structure has been changed. Information on the validation of tests used to assess body balance has been introduced with the corresponding summaries.

  1. Discussion – I think it would be better to state overall conclusion here…PA was effective in most but not all studies with limitations being… in other words, summarize the results. Ie: taken together, results of the systematic review suggest that….
  2. Could not go too deep into IGFBP’s and CRP (biology) as its pretty far from balance / postural data.
  3. Starting with ln 436- this is where you hone in on interpreting the results. Consider starting this discussion at this point.
  4. Ln 434 – there’s a misplaced z in this sentence.
  5. Ln 467- excellent comparison here. Can you make more comparisons among the other studies? Ln 474 – you do a great job of comparisons.
  6. Revisit paragraph structure in the discussion. There are usually 3 sentences minimum in each paragraph. You have the components here, but the discussion can be reformatted/ rearranged to allow for better flow.
  7. Ln 500 – can you spell out what high quality would be? Ample statistical power? Which balance test would you suggest to use, based on what you reviewed?